# Recyclable Thermoplastic Elastomer from Furan Functionalized Hairy Nanoparticles with Polystyrene Core and Polydimethylsiloxane Hairs

**DOI:** 10.3390/polym16223117

**Published:** 2024-11-07

**Authors:** Md Hanif Uddin, Sultan Alshali, Esam Alqurashi, Saber Alyoubi, Natalia Walters, Ishrat M. Khan

**Affiliations:** Department of Chemistry, Clark Atlanta University, Atlanta, GA 30314, USA; muddin@cau.edu (M.H.U.); sultan.alsahli@students.cau.edu (S.A.); ealqurashi@bu.edu.sa (E.A.); saber.alyoubi@students.cau.edu (S.A.); natalia.walters@students.cau.edu (N.W.)

**Keywords:** living anionic polymerization (LAP), thermoplastic elastomer, Diels–Alder reaction (DA), glass transition temperature (Tg), dynamic covalent bond, kinetics

## Abstract

Polymers synthesized with end-of-life consideration allow for recovery and reprocessing. “Living-anionic polymerization (LAP)” and hydrosilylation reaction were utilized to synthesize hair-end furan functionalized hairy nanoparticles (HNPs) with a hard polystyrene (PS) core and soft polydimethylsiloxane (PDMS) hairs via a one-pot approach. The synthesis was carried out by first preparing the living core through crosslinking styrene with divinylbenzene using sec-butyl lithium, followed by the addition of the hexamethylcyclotrisiloxane (D3) monomer to the living core. The living polymer was terminated by dimethylchlorosilane to obtain the HNPs with Si-H functional end groups. The furan functionalization was carried out by the hydrosilylation reaction between the Si-H of the functionalized HNP and 2-vinyl furan. Additionally, furan functionalized polystyrene (PS) and polydimethylsiloxane (PDMS) were also synthesized by LAP. ^1^H NMR and ATR-IR spectra confirmed the successful synthesis of the target polymers. Differential scanning calorimetry showed two glass transition temperatures indicative of a polydimethylsiloxane soft phase and a polystyrene hard phase, suggesting that the HNPs are microphase separated. The furan functionalized HNPs form thermo-reversible networks upon crosslinking with bismaleimide (BMI) via a Diels−Alder coupling reaction. The kinetics of the forward Diels–Alder reaction between the functionalized polymer and BMI were studied at three different temperatures: 50 °C, 60 °C, and 70 °C by UV–Vis spectroscopy. The activation energy for the furan functionalized HNPs reaction with the bismaleimide was lower compared to the furan functionalized polystyrene and polydimethylsiloxane linear polymers. The crosslinked polymer network formed from the Diels−Alder forward reaction dissociates at around 140–154 °C, and the HNPs are recovered. The recovered HNPs can be re-crosslinked at 50 °C. The results suggest that furan functionalized HNPs are promising building blocks for preparing thermo-reversible elastomeric networks.

## 1. Introduction

Plastic waste is a severe problem for the global economy. Municipal solid waste contains more than 10% synthetic polymers [1,2,3]. Over time, this will continue to increase as an environmental problem as societal demand continues to increase for polymers. One approach to alleviating this oncoming environmental concern is to develop polymers for multiple uses, i.e., recover and repurpose. Therefore, polymers synthesized with end-of-life considerations that allow for the recovery and reprocessing of the material are of significant interest. Polymers designed for recovery and reprocessing should be easy and cost-effective to synthesize. Diels–Alder (DA) chemistry is a powerful tool for studying the recovery and reprocessing of materials. DA reaction is a [4 + 2] cycloaddition reaction between an electron-rich diene and an electron-poor dienophile, which forms a DA adduct without any catalyst. It is a thermally reversible reaction that can be applied to various polymers [4,5,6,7,8]. It has been extensively studied due to the relatively fast kinetics and mild reaction conditions [8,9,10,11]. The crosslinking of polymers can be achieved by reacting polymers functionalized with furan (diene) and maleimide (dienophile) moieties. Moreover, its retro-DA reaction takes place at relatively low temperatures, which is the decrosslinking process. This, therefore, opens up exciting applications, such as recyclable networks and self-healing materials using functional polymers containing furans and maleimide. Polymer networks, polymer gels, and dendrimers have been prepared by reacting furan and maleimide functionalized polymers. The first report on this approach to preparing polymeric materials was by Stevens and Jenkins [12]. They used the Friedel–Crafts alkylation to functionalize polystyrene with a maleimide functional group and crosslinked the polystyrene using a bifunctional furan structure to create self-healing materials. A decade later, Canary and Stevens used this idea by mixing polystyrene functionalized by maleimide with di-furfuryl adipate [13]. Several groups have used furan as a diene in Diels–Alder reactions [14,15,16,17]. Bapat et al. and Barthel et al. have reported using a thermo-reversible Diels–Alder reaction to regulate the crosslinking of micelles [14,15]. Das et al. recently reported the formation of novel thermo-resettable thermosetting polymeric materials of furyl-modified copolymers (FMPs) and BMI using DA reactions [18]. Furthermore, Kang et al. used DA reactions for block copolymers that contain pendant triphenylamine [17]. The crosslinked network formed at 50 °C, which was confirmed by the DSC studies, which showed an increase in Tg, and for the decrosslinking reaction, a retro-Diels–Alder peak was observed at 149 °C in the DSC thermogram. Buonerba et al. reported the formation of the thermo-reversible crosslinked network via the Diels–Alder (DA) reaction of copolymers of 2-vinylfuran and styrene (S-co-2VFs) with BMI [19]. The temperature range for crosslinking and decrosslinking of S-co-2VFs with BMI via DA and r-DA reactions was between 50 °C and 150 °C. Recently, Liang et al. studied the kinetics of the forward Diels–Alder reaction between furfuryl-terminated polybutadiene (FTPB) and N, N0-1,3-phenylenedimaleimide (PDMI) by UV–Vis spectroscopy [20]. The reaction kinetics of FTPB-PDMI was measured by the change in the peak that appears at 310 nm because of the conjugated C=C–C=O moiety. The kinetics were studied at different temperatures, and it was found that the reaction rate increased with temperature.

Thermoplastic elastomers (TPEs) are an important class of materials and can be found in various high-value applications, such as luxurious footwear, catheters, tires, etc. They show rubber and plastic-like properties at room temperature due to their two glass transition temperatures, one higher than room temperature and another below room temperature. Hairy nanoparticles (HNPs) can be described as materials with an inner core and an outer hairy layer, sometimes called the shell. The combination of the characteristics and properties of core components and shell components in HNPs can show excellent properties that make them attractive [21,22]. Furthermore, because of their small size (10 nm to 100 nm), multiple functionalities on the shells can be included, and further, because HNPs have a high surface area-to-volume ratio, they have unique physical, chemical, and, sometimes, even biological properties. Therefore, they can be found as active materials for many applications in various areas of technology [23], such as catalysis [24], resistant coatings [25,26], pollution control, tires, the rubber industry [27], and sensing nanocarriers for drug and gene delivery [28,29]. Moreover, polymeric nanoparticles that have self-assembling properties have been used as building blocks for self-assembled nano-micro structures [30,31].

“Living anionic polymerization” (LAP) is a powerful synthetic method to synthesize well-defined polymers, block copolymers, and functional polymers [32]. Living anionic polymerization has been successfully used to synthesize well-defined thermoplastic elastomers (TPE). Zheng et al. synthesized TPE by grafting polybutadiene brushes to polystyrene core particles prepared by LAP (living anionic polymerization) [33]. Zhou et al. synthesized all-polymeric HNP with PDMS brushes and crosslinked polystyrene core by living anionic polymerization via a one-pot approach [34]. The synthesized HNPs were microphase-separated and acted like nanoscale TPE materials because two glass transition temperatures were observed.

In this communication, we report the synthesis of HNPs functionalized with the furan group by combining living anionic polymerization and hydrosilylation reactions. The furan group at the hair end makes the HNPs interesting as it can undergo crosslinking reaction with bismaleimide at low temperatures to form thermoplastic elastomeric networks, which can easily be de-crosslinked at a relatively low temperature of ~150 °C as shown in Figure 1.

## 2. Materials and Methods

Calcium hydride (CaH_2_), sec-butyllithium in cyclohexane (1.4 M), sodium metal, benzophenone, platinum (0), deuterated chloroform (CDCl_3_), 1,1′-(Methylenedi-4,1-phenylene) bismaleimide (BMI), nitrobenzene, anhydrous potassium carbonate, and methyl triphenyl phosphonium bromide were purchased from Millipore Sigma chemical company. Furfural, Styrene (S), divinylbenzene (DVB), hexamethylcyclotrisiloxane (D3), chlorodimethylsilane, and tetrahydrofuran (THF) were also purchased from Millipore Sigma and were purified before use. The procedures of purification are described below in detail. Methanol, 2-Propanol, and (2,2,6,6-Tetramethylpiperidin-1-yl) oxyl (TEMPO) were purchased from the Fisher Chemical Company. 2-Vinylfuran was synthesized in our lab, and a ^1^H NMR spectrum is provided to demonstrate purity (Appendix A). Speier’s catalyst (H_2_PtCl_6_) was purchased from Millipore Sigma (St. Louis, MO, USA). The hydrosilylation reactions were conducted under an inert nitrogen atmosphere. Tetrahydrofuran (THF) was purified by refluxing over sodium metal in the presence of benzophenone. A styrene and divinylbenzene mixture (90:10 S/DVB) was stirred overnight over finely powdered calcium hydride. On the vacuum line, the mixture was frozen–degassed–thawed three times before being distilled into pre-calibrated high-vacuum storage tubes. Styrene, hexamethylcyclotrisiloxane (D3), 2-vinylfuran, and the terminating agent chlorodimethylsilane were purified similarly.

Synthesis of Furan end-functionalized Hairy nanoparticles (HNP-Fs):

The HNPs functionalized with furan were synthesized by the high-vacuum living anionic polymerization technique. The high-vacuum storage tubes containing the mixture of a monomer and crosslinker (S and DVB), hexamethylcyclotrisiloxane (D3), chlorodimethylsilane, 2-vinylfuran, and tetrahydrofuran (THF) were attached in advance to the reactor. The reactor was placed under high vacuum and flushed several times with dry nitrogen. At first, 25 mL of dry distilled THF was transferred to the reactor from a high-vacuum storage tube. After that, the mixture of styrene (1.50 g) and divinylbenzene (0.22 g) was transferred to the reactor. The temperature of the monomer solution was reduced to −78 °C. Under vigorous stirring, 1 mL of sec-BuLi was rapidly added to the reactor by septum. The color of the reaction mixture immediately turned red, which is characteristic of the living styryllithium anion, indicating the occurrence of anionic polymerization. The first-step polymerization was carried out at −78 °C for 1.5 h. The second-step polymerization was initiated by the living styryllithium anion by adding 1.80 g of hexamethylcyclotrisiloxane (D3) dissolved in THF to the reaction mixture and gradually warming the reactor temperature to room temperature. The red color initially became orange and lightened over time until finally it disappeared completely. The second polymerization was then carried out at room temperature for 15 h. Finally, purified chlorodimethylsilane was added to the reactor from a high-vacuum storage tube to the reactor to terminate the polymerization. After 30 min, 2 mL of the reaction mixture was pulled by syringe and precipitated in methanol for ^1^H NMR and FTIR characterization before adding 0.3 gm 2-vinylfuran monomer to the reactor. Under continuous dry nitrogen flow to the reactor, 3–5 drops of Speier’s catalyst (H_2_PtCl_6_) and 5 mol% of TEMPO dissolved in dry THF were added, and the reaction was conducted for another 24 h at room temperature to complete the hydrosilylation reaction. Then, the resulting polymer was precipitated into a large excess of methanol. It was dried in a vacuum oven at room temperature to produce 3.5 g furan functionalized HNPs.

Thermo-reversible crosslinking via Diels–Alder Reaction:

The crosslinked TPE network was prepared by a Diels–Alder reaction between bismaleimide (BMI) and the furan functionalized HNP. The polymers were dissolved in N, N-dimethylformamide, and THF (1:1) into 50 mL round bottom flasks. The necessary amount of the BMI was then added to the polymer solution. The experimental details are given in Table 1. The mixture was stirred and placed into an oil bath set to 50 °C for 24 h. After that, the mixtures were precipitated into methanol to obtain the crosslinked yellow polymer. Then, the crosslinked polymer was dried under a vacuum at 50 °C and analyzed by DSC and ATR-IR. The DA reaction kinetics between all polymers and BMI were also studied at 50 °C, 60 °C, and 70 °C.

## 3. Results and Discussion

The hairy nanoparticles (HNPs) were synthesized from the sequential addition of the monomers (Figure 1). In the first step, the mixture of styrene and divinylbenzene (90 S: 10 DVB molar ratio) was polymerized in dry THF using sec-BuLi at −78 °C. In the second step, the cyclic hexamethylcyclotrisiloxane, D3, and monomer were added to the core and polymerized at room temperature. The living polymers were terminated by dimethylchlorosilane to obtain the HNP-SiH. The spectral data are consistent with published data [34]. The ATR-FTIR spectroscopy analysis of HNP-SiH in Figure 2 shows absorption bands of PDMS hairs and a characteristic absorption band of silyl hydride (Si-H) functionality at the end. A characteristic absorption band at 2120 cm^−1^ corresponds to a Si-H bond stretching vibration. The absorption band at 1261 cm^−1^ corresponds to the Si–CH_3_ group, whereas Si–O–Si anti-symmetrical stretching peaks are observed at 1097 and 1032 cm^−1^. The peaks at 758 and 846 cm^−1^ correspond to Si-C stretching. For the PS-core in HNP-SiH, the 2900–3000 cm^−1^ peaks correspond to the aliphatic C-H stretching, and the C-H aromatic peaks can be seen in the range of 3000–3090 cm^−1^. The peaks for the C=C aromatic are observed at 1453 cm^−1^, 1494 cm^−1^, and 1602 cm^−1^. There is a residual water/silanol (Si-OH) peak appearing around 3400 cm^−1^ due to the silylhydride (Si-H) group reacting with the water present in methanol during polymer precipitation. The ^1^H NMR spectrum shows the characteristic peak for the silane proton at 4.72 ppm. The characteristic signals at 0.25 to −0.25 ppm correspond to the PDMS brushes in HNP-SiH (Figure 3a). Alkyl protons of the backbone of the polystyrene (-CH_2_-CH-) cores appear in the range from 0.9 to 2.0 ppm, and the aryl protons from the phenyl ring show the characteristic peak ranging from 6.4 to 7.25 ppm. The methyl protons introduced from the sec-butyl group at the chain ends are observed at 0.74 ppm (Figure 3a). The range between 5.1 and 5.6 ppm shows no signals attributed to methylene protons (CH_2_=CH_2_) of unreacted vinyl groups stemming from the crosslinker DVB, indicating that all vinyl groups of the crosslinkers were copolymerized to form the crosslinked polystyrene core. The ratio of the SiH and PDMS chain was found to be 1:4.35 from proton signal integration. The styrene and the hexamethylcyclotrisiloxane (D3) monomers were also polymerized by the living anionic polymerization (LAP) technique using Sec-BuLi as an initiator and terminated by dimethylchlorosilane, according to Appendix A, to obtain silyl hydride terminated polystyrene and polydimethylsiloxane linear polymers. The functionalization of the materials was confirmed by the disappearance of the Si-H peaks and the appearance of new peaks that are characteristic of the furan in both ATR-FTIR and ^1^H NMR spectra (Figure 2, Figure 3, Appendix A). The new absorption peaks are observed at 1012 plus 1087 and 616 cm-1, corresponding to the furan heterocycle and the deformation vibration band of the furan ring, respectively. The ^1^H NMR spectrum of the HNP-Fs shown in Figure 3b confirms the disappearance of the Si-H peak at 4.54 ppm. The characteristic furan peaks appear at 5.6–6.4 ppm, which correspond to protons of the furan ring attached to the polymer—the methylene protons of furan overlap with the alkyl protons of the polystyrene backbone. No residual allyl protons between 5.1 and 5.6 ppm were detected in unreacted vinyl groups in furan. The proton signal integration of the furan (proton at position b) and PDMS chain ratio was found to be 1:5.6. This indicates that not all the silylhydride groups were functionalized after hydrosilylation reaction.

Dynamic light scattering (DLS) was used to measure the size of the HNP-SiH before and after furan functionalization by dissolving the respective samples in tetrahydrofuran. The measurement showed that the particles were in the nano-scale range. The radius of the HNP-SiH was found to be mostly between 40 and 55 nm (Figure 4a), whereas the radius of the HNP-F was found to be predominantly between the 50 and 70 nm (Figure 4b) range. The size difference in radius can be attributed to the aggregation of furan functionalized particles.

Thermogravimetric analysis (TGA) was used to study the thermal stability of all synthesized materials, and the onset temperature (T_d_), degradation maxima (T_max_), and % char yield are listed in Table 2. The analyses were performed using nitrogen flow and scanned from room temperature to 700 °C. Figure 5a shows the decomposition of PS-F (blue curve and indicated by blue single blue arrow), which begins at ~413 °C. The thermal decomposition of HNP-F (red curve) shows a two-step mass loss process between 25 °C and 700 °C, which indicates a two-phase system of hairy nanoparticles. The first degradation step, at approximately 382 °C (single red arrow), Figure 5a, is attributed to the decomposition of the PS Core. The second degradation step, at around 515 °C (double red arrow), corresponds to the decomposition of the PDMS hairs of the nanoparticles. When compared with the decomposition temperature for PS-F, the lowering of the decomposition temperature of the PS-core in the HNP is attributed to the interplay between the PS and PDMS segments.

Furthermore, DSC was also used to determine the thermal properties and the glass transition temperature (Tg) for all polymers, and the polymers’ thermograms are shown in Figure 5b. As seen in the DSC thermogram, the glass transition temperature of the PS was observed at 103 °C. The DSC thermogram of the HNP shows two distinct transition temperatures. The polystyrene core’s glass transition temperature (Tg) was observed at 110 °C. The higher glass transition temperature than the pure homopolymers is because the polystyrene core of the HNPs is crosslinked. Because of the operational limitations of the DSC instrument, a glass transition temperature (Tg) of the PDMS segment was not determined. However, a melting transition temperature (Tm) of the PDMS phase was observed at −24 °C. Therefore, the HNPs are microphase-separated materials and should function as thermoplastic elastomers [34].

Crosslinking via a Diels–Alder Reaction

The crosslinking of HNP-F was carried out by the reaction with a bismaleimide (BMI), i.e., a Diels–Alder reaction (Figure 2). The reaction between PDMS-F and PS-F and BMI was also studied. The FT-IR and UV–Vis spectrums were used to study the Diels–Alder reactions.

Figure 6 shows the visual observation of the crosslinking and decrosslinking of the HNP-F with BMI. Figure 6a is a solution of the HNP-F and BMI, and Figure 6b is the crosslinked gel prepared by heating to 50 °C for 2 h. The gel can be quickly decrosslinked (Figure 6c,d) by heating to 150 °C for one minute to recover the uncrosslinked HNP-F. The recovered HNP-F can be crosslinked again by heating it to 50 °C for two hours. The most interesting conclusion is how quickly and easily the decrosslinking reaction takes place, which strongly suggests the utilization of HNPs and building blocks for recoverable and reusable networks. The chemical reaction of the crosslinking can be followed by FT-IR spectroscopy. The absorption peaks in the range of 1190 cm^−1^ correspond to the characteristic succinimide {(CH2)2(CO)2NH} bands that result from the formation of the DA adduct, which indicates the crosslinking reaction (Figure 7) [35,36]. The peaks for C-O-C and Si-O-Si decrease upon crosslinking, as do the absorption bands at 616 cm^−1^, which relate to the furan ring. New absorption peaks are observed at 1720 and 1770 cm^−1^, corresponding to the BMI carbonyl groups [37,38].

Kinetics Study of the Diels–Alder Reaction by UV–Vis spectroscopy

The kinetics of the forward Diels–Alder reaction of all HNP-Fs, PSF, and PDMS-F with BMI were studied by UV–Vis spectroscopy at different temperatures (50 °C, 60 °C, and 70 °C). As seen in the UV–Vis spectra of the HNP-F, the maleimide absorption decreases with time at 310 nm. For HNP-Fs with BMI, the absorption at 310 nm dropped rapidly initially, indicating that maleimide groups reacted quickly (Figure 8A). After that, the absorption changed slightly only. On the other hand, the maleimide absorption of PS-MBI and PDMS-BMI decreased considerably slower than the HNP-F (Appendix A).

The degree of the reaction with the reaction time was calculated using the following expressions:(1)$$\frac{dx}{dt}=k{\left(1-x\right)}^{n}$$
where x is the conversion rate, which can be expressed as
(2)$$x=1-\frac{A_{t}}{A_{0}}$$

By rearranging Equation (2), we have
(3)$$1-x=\frac{A_{t}}{A_{0}}$$
where *A*_0_ is the initial absorbance at (t = 0), and *A*_t_ is the absorbance at 310 nm at time *t*.

When the reaction is first-order (n = 1), Equation (1) can be written as for the pseudo-first-order reaction:(4)$$\ln\frac{1}{1-x}=k_{1}t$$

When the reaction is second-order (n = 2), Equation (1) can be written as
(5)$$\frac{1}{1-x}=k_{2}t$$

The reaction rate constants for the first- and second-order (*k*_1_, *k*_2_) for the DA reaction were calculated for each temperature and are listed in Table 3. The rate constants were derived from the slope of the graphs shown in (Figure 7). The reaction rate constants (*k*_1_, *k*_2_) increased when the temperature rose from 50 °C to 70 °C. This would significantly accelerate the crosslinking reaction rate because the rate constants’ value significantly changes with temperature. When comparing all the results of order kinetics, the DA reaction between the polymers and BMI followed second-order kinetics [39,40,41].

The DA reactions’ activation energy was calculated using the Arrhenius plot.

Arrhenius equation:(6)$$\ln k=-\frac{E_{a}}{RT}+\ln A$$

*E_a_* is the activation energy, *R* is the universal gas constant (R = 8.314 J mol^−1^ K^−1^), *T* is the absolute temperature (K), and *A* is the pre-exponential factor. The DA reaction activation energy was calculated from the rate constants of the second-order reaction kinetics at 50 °C, 60 °C, and 70 °C, which was determined by the slope of the line from graphing ln *k* versus 1/T, shown in Figure 8D. The activation energy of the crosslinking reaction for the HNP, PS, and PDMS with BMI was 36.91, 45.31, and 41.34 kJ mol^−1^, respectively, and they are listed in Table 4. The values are similar to those found in previous studies for the DA reaction in polymeric systems [36,39,40]. Compared to the three crosslinking reactions of HNP, PS, and PDMS, the *Ea* of the Diels–Alder reaction of HNP with BMI is smaller than furan functionalized homopolymers. This result may be because of the HNP-F’s spherical particle shape, which may be an approach to control assembly and disassembly processes. Furthermore, the chain flexibility of PDMS-F resulted in a lower *E_a_* compared to the more rigid one of the PS-F chain in the crosslinking reaction of homopolymers.

Differential scanning calorimetry (DSC) was used to study the retro Diels–Alder reaction of the crosslinked polymers with BMI. Figure 9a–c shows the heating curves of the DSC analysis of HNP-F, PS-F, and PDMS-F after reaction with the BMI. As seen in the thermogram, a broad peak at 154 °C is observed, indicating the cleavage of HNP-BMI via a retro-DA (rDA) reaction in the first heating curve. No peak is observed in the second heating curve since the cooling process time was not long enough to form the DA adduct [37].

Scanning electron microscopy was used to investigate the morphology of the HNP-F, PS-F, and PDMS-F before and after the crosslinking reaction with BMI. All samples were prepared by drop-casting on silicon wafers using chloroform as the solvent. The crosslinked polymers showed different morphologies. Crosslinked HNP-F TPEs showed the most uniform surface with some anisotropic pores (Figure 10b), which is expected because the available polymer did not totally cover the silicon wafer surface. The SEM image (Figure 10c), decrosslinked after heating the silicon wafer to 150 °C, does not have surface smoothness. The difference between the two uncrosslinked SEM images (Figure 10a,c) can be attributed to the preparation of the two samples, i.e., one was drop-cast (Figure 10a), and the second one (Figure 10c) was heated first to 50 °C and then to 150 °C to decrosslink it. Furthermore, AFM images of crosslinked HNP-F also reveal smooth porous surfaces (Figure 10d,e).

## 4. Conclusions

In summary, we synthesized chain-end furan functionalized hairy nanoparticles (HNP-Fs) with a hard polystyrene (PS) core and soft polydimethylsiloxane (PDMS) shells via a one-pot approach. The furan functionalized HNPs were synthesized, combining sequential living anionic polymerization and hydrosilylation reaction. Differential scanning calorimetry showed the presence of two thermal transitions, indicative of the presence of a polydimethylsiloxane soft phase and a polystyrene hard phase, suggesting that the HNPs are a thermoplastic elastomer. Additionally, furan-functionalized PS and PDMS were successfully synthesized and reacted with bis-maleimide (BMI). The activation energy of the forward Diels–Alder reaction of BMI with HNP-F, PS-F, and PDMS-F was determined to be 46.57, 67.61, and 54.22 kJ mol^−1^, respectively. These results indicated that the activation energy (*E_a_*) was highly dependent on the polymer shape and flexibility. HNP-F, due to its shape, flexibility of hairs, and uniformity, needs less energy to react with BMI compared to linear polymers. Moreover, SEM and AFM images show uniform crosslinked surfaces originating from HNP-F. BMI can crosslink the HNP-F by heating it to 50 °C and decrosslink it by heating to 150 °C for one minute to recover the uncrosslinked HNP-F. The recovered HNP-F can be crosslinked again by heating it to 50 °C for two hours. The most interesting conclusion is how quickly and easily the decrosslinking reaction takes place, and it strongly suggests the utilization of HNPs as building blocks for recoverable and reusable networks. The study indicates furan functionalized HNPs are promising TPE building blocks for facile recovery. The HNP-F can be easily reprocessed and potentially minimize thermoplastic elastomeric waste in the environment.

## Data Availability

Data supporting this study are included within the article.

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
