# Peer review of "Recyclable Thermoplastic Elastomer from Furan Functionalized Hairy Nanoparticles with Polystyrene Core and Polydimethylsiloxane Hairs"

_polymers, 2024, doi:10.3390/polym16223117_

Round 1

Reviewer 1 Report

Comments and Suggestions for Authors

This manuscript is devoted to the preparation of hairy nanoparticles based on polystyrene and polydimethylsiloxane. The manuscripts utilized various reliable techniques, including DSC, TG, NMR, FTIR, and others. However, there are several concerns related to the quality of presentation and scientific soundness, listed below.

Content questions:

1)    The title can be more specific.

2)    In Figure 2 all signals discussed in the main text should be labeled or marked. In addition, what is the reason for changing in relative intensities of PDMS bonds at 1261 cm-1 and 1097-1032 cm−1 corresponding to HNP-SiH and HNP-F.

3)    In the NMR spectrum (Figure 3), more signals can be labeled. For example, alkyl protons of the polystyrene backbone. The description of the furan-functionalized HNP is strange - The characteristic furan peaks appeared at 5.5-6.4 ppm instead of 6.1-6.4 ppm. What is the nature of a huge singlet around 1.5 ppm on HNP-SiH? In addition, the degree of furan-functionalization can be checked via integral intensities. For example, using the ration of PDMS signals (from 0.25 to -0.25 ppm) to signal a in both HNP-SiH and HNP-F.

4)    The melting point of PDMS reported as -41 degC (Fig.4 II) also appeared in the PS-F. Does it mean that it is an instrumental error?

5)    The scale in Figure 6 is unreadable.

6)    The precision of the results reported in Table 2 and Table 3 is very strange.

7)    Does the color appear again when HNP-F decrosslinking after being heated at 150 degC?

8)    There is no evidence that the particles are nanosized.

Technical questions:

1)    Upper script for units should be used (for example, cm−1 in lines 182, 183, 280).

2)    A lot of missing space (line 188), especially between number and units (line 150, 153,…)

3)    Extra symbols (line 140, vacu-um; or-ange line 149; polymer-ization 150)

4)    Abbreviations (for example, twice introduce styrene (S) – lines 119 and 175; abbreviations of HNP-F and HNP-SiH).

5)    Duplication of concentrations (from part 2 to part 3, like sec-BuLi (1.4 M) – lines 115, 142, and 176)

6)    Multiple standards for similar things – for example, labeling figures in Fig.4 as I and II, while in Fig.3 labels are a) and b).

7)    AFM images should keep an aspect ratio when resizing.

Reviewer 2 Report

Comments and Suggestions for Authors

The article considers Hairy nanoparticles (HNP-F) functionalized with furan. In the introduction, the authors pointed out the relevance of the synthesis and subsequent use of such compounds. Nevertheless, a number of both significant and minor questions and comments arise regarding the presented work:

1. The authors should work significantly on the design of the manuscript:

• The list of references, for example, should be brought into line with the requirements of the journal. In such a scattered form, many of them do not even have a title so that they can be found and related to the topic.

• Figure captions and subtitles should also be designed accordingly. Somewhere there are indices (i) and (ii) to designate different parts of the figure instead of the standard (a) and (b), and the curves have letter markings instead of numeric ones. It is also relevant to use color marking of the curves with the corresponding decoding in the figure caption.

• All figures have different font types and sizes (especially Figs. 2 and 6). This should be corrected.

• In several places, the value -1 must be made in the superscript (Table 3, line 320, line 377).

• The "suns" in schemes 1, 2 and Fig. 3 have different formats. Is this intended or are they flattened somewhere? I think they should be made the same.

• Fig. 8 has strange quality and looks very much like a scan.

• There are strange signs when referring to figures (for example, lines 365 and 366). This should be corrected.

2. The text of the article and the supplementary materials do not contain information about the processing of IR spectra and the software used. Without information about the processing (including normalization, for example), it is difficult to compare the spectra with each other.

3. The authors indicated that chlorodimethylsilane was added to stop the polymerization. Are there any options for using another component? How preferable is it to use chlorodimethylsilane for the subsequent addition of 2-Vinylfuran? How did the presence of chlorodimethylsilane at the ends of the PS polymer chains affect the NMR and IR spectra?

4. Figure S2 contains IR spectra of the polymers at different stages of the synthesis: 1) the green spectrum is for some reason not normalized to the baseline compared to the others. 2) the blue spectrum in the middle region looks strange - one of the peaks is clearly a different color. It is necessary to provide this figure in better quality and in a uniform format.

5. In Scheme 1, during the addition of both chlorodimethylsilane and 2-Vinylfuran, one fragment is shown at the ends of the HNP hairs. According to the authors, how many hairs from the entire particle participate in this process? How many furan groups are formed on HNP? It is necessary to add this to the text.

6. Fig. 2 shows strange spectra. The spectrum for HNP-SiH (a) contains an abundance of benzene rings, which disappear somewhere in the spectrum of HNP-F (b). The authors actually described the absorption peaks of the PS core, but it is unclear why all this is not reflected in the HNP-F spectrum. Where does the wide absorption peak of hydroxyl groups in the HNP-SiH spectrum come from, which disappears upon transition to the HNP-F spectrum? These spectra should be described in more detail.

7. Fig. 4 – Why don’t the authors make a single color designation of the curves for the TGA and DSC data, and it is necessary to put in the figure caption which color corresponds to which object. And also carefully check the reference to these figures in the text. What do the arrows near the curves mean? Different stages of HNP-F thermal decomposition? This should be indicated in the text.

8. Fig. 6 – If HNP-F transitions from a cross-linked to a non-cross-linked state at different temperatures (and quickly), then how exactly was the IR spectroscopy carried out? Is the spectrum of non-cross-linked HNP-F shown the same as in Fig. 2? Have the authors tried to compare the spectra of specific mixtures before and after crosslinking? Since the spectrum of decrosslinked HNP-F obtained earlier was used, and not the one that was crosslinked, the attached IR spectra differ significantly from each other. The authors indicate only one small absorption peak to confirm their conclusions. Here, for example, hydroxyl groups are again present in large quantities (the spectrum of crosslinked HNP-F), which again remains unexplained. Moreover, the scale of the IR spectra does not allow for the precise determination of the positions of individual interesting bands, and the authors do not describe them at all. All IR spectroscopy data must be significantly reworked by adding a detailed description of the changes that occur.

9. For Figures 7 and S4, data on the error in determination are not presented, as well as how this affects subsequent calculations.

10. Fig. 9 – Why does the SEM image of decrosslinked HNP-F have such a fibrous structure? This does not correlate very well with the data presented by the authors in scheme 1. A description of the SEM data with a specific indication of what is what in the microphotograph should be added to the text. And similarly, it is necessary to indicate on other SEM images what has changed after crosslinking and subsequent decrosslinking.

Reviewer 3 Report

Comments and Suggestions for Authors

Review Report _ polymers-3255307

I have gone through the article entitled “Functionalized Hairy Nanoparticles for Reversible Thermoplastic Elastomeric Networks”. The authors have done very good work. However, there is the need to clarify and revise the following points before its acceptance.

1.      Please mention the full form of TEMPO where it has appeared first.

2.      Please include also DTG graph in the result and discussion section (Page No. 7, Figure 4). This will provide clear degradation maxima for the materials.

3.      The weight loss for PDMS-F has reached below 0wt% after 600 °C, that means showing negative value. Please explain this.

4.      Mention the important peak position values within FTIR figure. This will help the readers to easily identify the peaks.

5.      The authors have mentioned that “However, a melting transition temperature (Tm) of the PDMS phase was observed at -41 °C.” In Figure, it is looking that the like Tm is around -25 °C. Please clarify this point.

6.      It is better if the authors mention the thermal properties like onset of degradation, degradation maxima, %residue, Tg, and Tm within a table.

Round 2

Reviewer 1 Report

Comments and Suggestions for Authors

Thank you for the scientific answers and comments.

Reviewer 2 Report

Comments and Suggestions for Authors

The authors have worked very well on the manuscript and answered all the questions. The article can be accepted in this form.

A small remark - in my version of the file with corrections, some of the corrected figures overlapped the old version of the figures a little. But I think that all this will be corrected during the proofreading.